# Metabolic Profiling and Molecular Mechanisms Underlying Melatonin-Induced Secondary Metabolism of Postharvest Goji Berry (*Lycium barbarum* L.)

**DOI:** 10.3390/foods12234326

**Published:** 2023-11-29

**Authors:** Junjie Wang, Huaiyu Zhang, Jie Hou, En Yang, Lunaike Zhao, Yueli Zhou, Wenping Ma, Danmei Ma, Jiayi Li

**Affiliations:** 1Key Laboratory of Storage and Processing of Plant Agro-Products, School of Biological Science and Engineering, North Minzu University, Yinchuan 750021, China; smkxwjj@163.com (J.W.); domybest-001@163.com (H.Z.); hj090703@163.com (J.H.); yangen1213@163.com (E.Y.); zhao921526641@163.com (L.Z.); 15732950990@163.com (Y.Z.); 19948736865@163.com (D.M.); 18395098202@163.com (J.L.); 2School of Food and Biological Engineering, Hefei University of Technology, Hefei 230601, China

**Keywords:** *Lycium barbarum* L., melatonin, phenylpropanoid pathway, antioxidant enzyme, secondary products

## Abstract

Postharvest decay of goji berries, mainly caused by *Alternaria alternata*, results in significant economic losses. To investigate the effects of melatonin (MLT) on resistance to *Alternaria* rot in goji berries, the fruits were immersed in the MLT solutions with varying concentrations (0, 25, 50, and 75 μmol L^−1^) and then inoculated with *A. alternata*. The results showed that the fruits treated with 50 μmol L^−1^ MLT exhibited the lowest disease incidence and least lesion diameter. Meanwhile, endogenous MLT in the fruits treated with 50 μmol L^−1^ MLT showed higher levels than in the control fruits during storage at 4 ± 0.5 °C. Further, the enzymatic activities and expressions of genes encoding peroxidase, phenylalanine ammonia-lyase, cinnamate 4-hydroxylase, 4-coumarate-CoA ligase, chalcone synthase, chalcone isomerase, and cinnamyl alcohol dehydrogenase were induced in the treated fruit during storage. UPLC-ESI-MS/MS revealed that secondary metabolites in the fruits on day 0, in order of highest to lowest levels, were rutin, *p*-coumaric acid, chlorogenic acid, ferulic acid, caffeic acid, naringenin, quercetin, kaempferol, and protocatechuic acid. MLT-treated fruits exhibited higher levels of secondary metabolites than the control. In conclusion, MLT treatment contributed to controlling the postharvest decay of goji fruit during storage by boosting endogenous MLT levels, thus activating the antioxidant system and secondary metabolism.

## 1. Introduction

Goji (*Lycium barbarum* L.) fruit is a traditional good source of homology of medicine and food in China [1]. Goji berry has recently been categorized as the latest “superfood” or “berry of youth”, owing to its high nutritional value and a variety of pharmacological functions, which generally also depend on genotype [2,3,4]. It contains vitamins, minerals, microelements, polyphenols, polysaccharides, and carotenoids and exhibits biological functions including immunomodulatory, anti-defying, anti-oxidative, anti-inflammatory, anti-cancer, and resist fatigue [5,6,7], making it globally popular. Fresh goji fruit is difficult to store, transport, and market during the mature period (early June to late September) because of the high surrounding temperature and the tender and juicy texture of the fruit, making it prone to mildew and decay [8,9]. Although fresh fruit is healthier and more nutritious, goji is usually circulated in dried or processed forms. Drying or processing of the fruit often leads to losing nutrients and bioactive compounds [5,10]. Therefore, the goji berry industry requires a new, eco-friendly method to preserve freshness and extend postharvest life.

Traditionally, goji fruit preservation depends on applying physical and chemical approaches, such as low-temperature handling and chemical fungicides [5,9]. In recent years, some safe and environment-friendly methods for the postharvest handling of goji berries, including lectin dipping [2], salicylic acid treatment [7,8], H_2_S fumigation [9], cysteine treatment [3], plasma-activated water immersion [10], chitosan coating [11], and acidic electrolyzed water treatment and vacuum precooling [12], have also been used, documented, and reported to inhibit rot, preserve the quality of the fruit, and increase its shelf life. Nevertheless, exploring novel preservation strategies for fresh goji fruit is still an urgent requirement.

Melatonin, also known as N-acetyl-5-methoxytryptamine (MLT), is a molecule found in living organisms that has a variety of effects [13]. In plants, MLT participates in various physiological processes related to growth and development, such as seed germination, rooting induction, flowering, photosynthesis, ripening, and senescence. MLT also behaves as an antioxidant-scavenged reactive oxygen species (ROS) or reactive nitrogen species (RNS) [14,15,16]. Furthermore, it tolerates many biotic and abiotic stresses, including pathogens, heat, cold, drought, salinity, alkalinity, heavy metals, and UV radiation [13,17,18,19]. Recently, exogenous MLT application has been shown to promote endogenous MLT levels, thus modulating different physiological changes in fruit [15,20]. MLT treatments can delay postharvest senescence and maintain the fruit quality of sweet cherries and peaches by enhancing antioxidant activity [21,22]. Fruits stored at low temperatures often suffer serious chilling injury. However, low-temperature storage enhances cold tolerances on plum fruit [23] and litchi fruit [24] by modulating secondary metabolism and energy supply metabolism. More recently, Arnao and Hernández-Ruiz [15] reported that MLT acts against pathogenic fungal or bacterial attacks in plants, playing a crucial role in mediating the disease response comprising ethylene, abscisic acid, salicylic acid, jasmonic acid, ROS metabolism, and secondary metabolism. Previous research has shown that the application of MLT can regulate jasmonic acid signaling and the phenylpropanoid pathway to inhibit the postharvest rot of blueberry fruit and induce signaling molecules via secondary metabolism to inhibit the growth of gray mold on cherry tomato fruit [17,25].

Phenylpropane-derived secondary metabolites, including total phenols, flavonoids, and lignin, play critical roles in plant development, stress resistance, and defense response. These compounds are abundant in young plant tissues, protecting young fruit. With ripening and senescence, the levels of these metabolites gradually decrease, and defensive resistance against stresses and pathogens becomes weaker [25,26]. Secondary metabolites are primarily synthesized and regulated by phenylalanine ammonia-lyase (PAL), cinnamate 4-hydroxylase (C4H), 4-coumarate-CoA ligase (4CL), chalcone synthase (CHS), chalcone isomerase (CHI), and cinnamyl alcohol dehydrogenase (CAD) of the phenylpropanoid pathway, and peroxidase (POD) [7]. Further, the secondary metabolism can be induced by the exogenous elicitors (such as salicylic acid, methyl jasmonate, and MLT) and biotic and abiotic stresses, and accumulating these metabolites enhances plant defense [17,26,27]. Our previous works have revealed the effectiveness of MLT treatment on retaining the postharvest goji fruit quality [28]. However, there have been no reports on the influence of MLT on postharvest goji berry decay and the mechanisms underlying it. Therefore, this study aims to determine the impact of MLT on goji berry decay induced by *Alternaria alternata* during storage and explore the MLT-mediated modulation of secondary metabolism and metabolic profiling to identify the potential underlying defense mechanisms.

## 2. Materials and Methods

### 2.1. Plant Materials

Goji berries (*Lycium barbarum* L. cv. Ningqi No. 5) were harvested by hand with pedicels from a local farm located in Yinchuan, China at the Ningxia Academy of Agriculture and Forestry Sciences in June 2018. Within 1 h postharvest, the fruits were transported to the laboratory in plastic boxes containing 100 each. The chosen fruits were consistent in color and size, with no noticeable imperfections. They were divided into four groups and immersed in MTL solutions with varying MLT concentrations of 25, 50, and 75 μmol L^−1^ and distilled water (control) for 5 min. A total of 300 mL of solution was used to treat 30 fruits, with 300 fruits per treatment. The dipped fruits were then air dried at 25 ± 0.5 °C within 60 min. The air-dried fruits were collected in plastic boxes (100 fruits per box; 15 × 10.5 × 6.5 cm^3^) and stored at 4 ± 0.5 °C (relative humidity: 85 ± 5%) for 30 d. The stored fruits were cut into pieces, and the seeds were removed for preparing the samples on days 6, 12, 18, 24, and 30 after storage. The prepared fruit samples were immediately frozen rapidly in liquid nitrogen and stored at −80 °C for subsequent biochemical and gene analysis.

### 2.2. Injury Inoculation and Investigation of Disease Expansion

*A. alternata* was isolated from *L. bararum* L. fruit with typical symptoms of *Alternaria* rot. Then, spore suspension preparation and inoculation tests were executed [8]. The spore suspension of the fungi was diluted to a total spore concentration of 1 × 10^6^ spores mL^−1^. On day 0, the stored untreated and treated fruits were wiped with 75% ethanol for surface disinfection (10 fruits per replicate per treatment). A uniform hole (2 mm deep, 1 mm wide) was made on the equator of each fruit using a sterile needle, followed by injection of the spore suspension (2 μL) into each wound. After air-drying for 60 min, the inoculated fruits were stored at 28 °C (relative humidity: 80–90%) for 3 d. Then, the disease incidence and lesion diameter were evaluated every day.

### 2.3. Analysis of Endogenous MLT Levels

Endogenous MLT content was determined using an MLT-based ELISA kit following the instructions of the kit (Renjie Biotechnology Co., Ltd., Shanghai, China). The MLT content was expressed as picograms per gram of fresh weight (pg/g FW).

### 2.4. Assay of Activities of Defense-related Enzymes

POD activity was measured using the methods previously described by Li et al. [17]. The reaction system consists of 0.1 mL of crude enzyme extract and 2.9 mL of phosphate buffer containing 30 mmol L^−1^ H_2_O_2_ and 25 mmol L^−1^ guaiacol. The mixture was then observed for 3 min, during which the oxidation of guaiacol to tetraguaiacol was detected at 470 nm using a spectrophotometer. One unit (U) was defined as a change in A_470_ per minute.

PAL, 4CL, C4H, and CAD were extractions using methods previously described by Zhang et al. [7]. The PAL reaction system comprised enzymatic extracts (100 μL) and Tris-HCl (3 mL; pH 8.9) containing 7 mmol L^−1^ L-phenylalanine. One U of PAL was defined as an increase in A_290_ of 0.01 per hour. The reaction solution of C4H consisted of enzymatic extracts (100 μL) and 50 mmol L^−1^ phosphate buffer (1.9 mL; pH 7.5) containing 0.5 mmol L^−1^ NADPH and 2 mmol L^−1^ trans-cinnamic acid. One U of C4H was equal to the change in A_290_ of 0.1 per hour. Crude enzyme solution of 4CL (100 μL) was mixed with a 3 mL reaction solution containing 0.2 mmol L^−1^ 4-coumaric acid, 2.5 mmol L^−1^ MgCl_2_, 50 μmol L^−1^ ATP, and 20 μmol L^−1^ coenzyme A. The system was kept at 37 °C for 0.5 h. One U of 4CL was calculated as an increase in A_333_ of 0.01 per hour. For CAD activity, 100 μL enzymatic extract solution was added to 0.4 mL of 0.1 mol L^−1^ phosphate buffer (pH 6.3) containing 5 mmol L^−1^ of trans-cinnamic acid and 10 mmol L^−1^ NADP+. One U of CAD activity was a change in A_340_ of 0.01 per hour. To determine the CHS activity, an enzyme-linked immune assay was performed using the commercially available CHS assay kit (JiangLai, Shanghai, China). To measure the CHI activity, a CHI assay system kit (Comin, Suzhou, China) was used. All assay procedures were executed as described by manufacturers’ protocols. CHI and CHS activities were defined as U mg^−1^ protein and nmol L^−1^, respectively.

All the results on enzymatic activity were expressed as U mg^−1^ protein. The concentrations of protein were determined using the same method as described previously by Bradford [29].

### 2.5. Expression Analysis of Genes Related to the Phenylpropanoid Pathway

Fruit tissue of goji berries was used for total RNA extraction using a plant total RNA extraction kit (Tiangen, Beijing, China). The total RNA underwent reverse transcription, wherein cDNA was generated using TransScript^®^ One-Step gDNA Removal and cDNA Synthesis SuperMix (Transgen, Beijing, China). The synthesized cDNA served as a template for quantitative real-time PCR (qRT-PCR) using SYBR Green Master qPCR Mix (Thermo, Waltham, MA, USA) with Actin and EF1α employed as internal references. The qPCR was performed on a LightCycler 480II RT-PCR system (Roche, Basel, Switzerland) [7]. To start the PCR protocol, there was a pre-denaturation step of 5 min at 95 °C. Then, amplification was carried out for 40 cycles at the following temperatures: 95 °C for 15 s, 65 °C for 30 s, and 72 °C for 30 s. The primers used for the process are listed in Table 1. To determine the relative expression of genes related to the phenylpropanoid pathway, the 2^−∆∆CT^ method was used [30].

### 2.6. Analysis of Secondary Metabolites

Samples from the untreated and treated (50 μmol L^−1^ MLT) goji fruit were extracted and detected [7]. A total of 200 mg freeze-dried samples were ground into powder and then extracted twice with aqueous methanol. The extracts were dried and re-dissolved in aqueous methanol containing L-2-chlorophenylalanine (12 ng mL^−1^) before being centrifuged at 4 °C and 12,000× *g*. The ultraperformance liquid chromatography–electrospray ionization–tandem mass spectrometry (UPLC-ESI-MS/MS) detected the secondary metabolites. The UPLC system used was ExionLC AD and the MS system used was QTrap 6500^+^. Both systems were developed by SCIEX, Framingham, MA, USA, and ESI sources. The analysis used an Analyst 1.7.1 workstation (SCIEX, Framingham, USA). Using a C18 column with 1.8 μm particle size, 2.1 mm by 100 mm (Waters Acquity UPLC HSS T3, Milford, CT, USA), and a 5 μL injection volume, phenolics and flavonoids were separated. The mobile phases used in the experiment consisted of two components: water with 0.1% formic acid (A) and acetonitrile (B). The gradient program involved a series of proportions, which were as follows: 98:2 (*v*/*v*, A:B) for 1 min; 80:20 for 2.5 min; 65:35 for 8.5 min; 45:55 for 10.5 min; 2:98 for 10.8 min; 2:98 for 12 min; 98:2 for 12.01 min; and 98:2 for 14 min. To analyze the metabolites, the MS was operated in a multi-reaction detection mode (MRM) under both positive and negative ion modes, and its parameters, including the declustering potential (DP) and collision energy (CE), were optimized. During the MS analysis, the following optimized conditions were applied: collision gas (35), ion spray voltage (+) 5500 V/(−) 4500 V, ion spray temperature (600 °C), and ion source gas (gas1: 45, gas2: 50). To ensure the repeatability of the results, quality control samples were injected every 10 samples throughout the analytical run. These samples provided additional data for assessing the accuracy and consistency of the analysis.

### 2.7. Statistical Analysis

The experiments were repeated at least three times to ensure accuracy. The results were presented as the average value and standard deviation (SD). Statistical analysis was performed using SPASS 19.0 software, with ANOVA and Duncan’s multi-range test utilized to determine significant differences (*p* < 0.05) between the means.

## 3. Results

First, we screened the most effective MLT treatment by exposing the fruits to solutions with varying MLT concentrations (25, 50, and 75 μmol L^−1^) and assessing disease severity. Then, the fruits treated with optimum MLT concentration were assessed for enzyme activity and gene expression and subjected to metabolic profiling.

### 3.1. Disease Incidence, Lesion Development, and Endogenous MLT Levels

Disease incidence and lesion diameter rapidly increased during storage in all untreated and treated fruits. All MLT-treated fruits exhibited marked inhibition of *A. alternata*-induced disease development, with the most prominent disease suppression observed in fruits treated with 50 and 75 μmol L^−1^ MLT. Compared with the untreated group, from day 1 to day 3, fruits treated with 50 and 75 μmol L^−1^ MLT solutions exhibited 42.86% and 39.29% lower disease incidence, while fruits treated with 25, 50, and 75 μmol L^−1^ MLT solutions exhibited 26.16%, 42.32%, and 10.32% decrease in lesion expansion (Figure 1A,B). Figure 1C illustrates that the control group exhibited visible disease spots on the second day following inoculation. By the third day, the symptoms had worsened significantly. In contrast, the 50 μmol L^−1^ MLT treatment demonstrated signs of black mold on the third day. Notably, exposure to 50 μmol L^−1^ MLT was the most effective of the three treatments selected for this study. Therefore, all the following experiments were conducted on fruits treated with 50 μmol L^−1^ of MLT.

The control fruits showed a decrease in MLT content for the first 6 days, as seen in Figure 2. After that, the content increased rapidly and peaked on day 18, followed by another decrease. On the other hand, the treated group displayed an increase in MLT content, reaching its highest value on day 12, and then gradually decreasing. During storage, the treated berries consistently had higher MLT content than the control group, with the former showing 5.21-fold higher MLT content on day 12.

### 3.2. Activities of Defense-Related Enzymes

As shown in Figure 3A,E,G, POD, CHS, and CAD activities constantly increased in the treated and untreated groups during the entire storage period. The treated berries exhibited markedly higher POD, CHS, and CAD activities (101.75%, 24.06%, and 34.05%, respectively) than the control group on day 18. Further, the treated group exhibited a gradual increase in PAL activity, unlike the control fruits, which showed a declining trend. PAL activity was dramatically higher in the treated berries (1.84 fold) than in the untreated berries on day 24 (Figure 3B). C4H activity gradually increased in the two groups until day 12 and subsequently rapidly decreased. However, the treated group significantly decreased the rate of decline in C4H activity during storage (Figure 3C). 4CL activity rapidly reduced in the untreated and treated fruits until day 12, then increased and reached a peak value on day 18, followed by a sharp decrease in the untreated fruit until day 30. The curves of CHI activity for all groups showed two peaks during storage. The treated fruits exhibited higher 4CL (1.22 fold, Figure 3D) and CHI (1.04 fold, Figure 3F) activities than the untreated fruit on day 30, respectively.

### 3.3. Expressions of Genes Encoding Defense-Related Enzymes

The gene expressions of *PAL* and *CHS* in the control group exhibited a constant decreasing trend during storage. However, from day 12 to 30, MLT-treated fruits exhibited significantly higher *PAL* and *CHS* expressions (both ≥ 4 fold) compared to the control on day 18, respectively (Figure 4A,D). As depicted in Figure 4B, the *C4H* transcript levels in the untreated group decreased until day 6, reached a maximum on day 12, and rapidly declined from day 12 to 24, followed by a rise afterward. The treated fruits exhibited significant *C4H* upregulation during storage. The relative expressions of *4CL* and *CAD* in the control fruit fluctuated with two peaks on days 6 and 18. The treated fruits exhibited prominent *4CL* and *CAD* upregulation, with 1.95- and 1.28-fold higher expressions on days 12 and 18, respectively, than the control (Figure 4C,F). Finally, in the untreated group, the *CHI* expression reduced until the first 12 days, increased from day 12 to day 24, and then rapidly declined. However, the treated fruits exhibited significant *CHI* upregulation compared to the untreated group, with more than 2-fold higher *CHI* expression on day 18 (Figure 4E).

### 3.4. Metabolic Profiling of Secondary Metabolite Pathway

Ten secondary metabolites in the untreated and treated fruits were investigated using UPLC-ESI-MS/MS during storage. As depicted in Figure 5A,B,D,F,I, the contents of caffeic acid, chlorogenic acid, *p*-coumaric acid, protocatechuic acid, and rutin showed rapidly declining trends in all groups throughout the storage period, except the slight rise of rutin content in the treated group after day 18. MLT treatment significantly mitigated the decrease in the levels of the metabolites mentioned above. The treated group exhibited 97.00, 138.92, 59.71, 150.00, and 16.51% higher levels of caffeic acid, chlorogenic acid, *p*-coumaric acid, protocatechuic acid, and rutin than the untreated group on day 30, respectively. Furthermore, the levels of ferulic acid and sinapic acid for both treated and untreated groups decreased during the first 6 days and then increased from day 6 to day 18 (Figure 5C,E). After day 18, ferulic acid levels showed a single peak trend while the levels of sinapic acid gradually rose. Relative to the control group, the treated group maintained higher ferulic acid levels during the entire storage period, with more than 2-fold higher levels on day 30. However, the sinapic acid contents were not significantly different between the treated and untreated groups. As shown in Figure 5G,H,J, the contents of the naringenin, quercetin, and kaempferol in both treated and untreated groups, exhibited a single peak on day 18. Moreover, MLT treatment promoted the accumulation of naringenin, quercetin, and kaempferol, as evidenced by 69.38%, 103.92%, and 88.12% higher levels of naringenin, quercetin, and kaempferol, respectively, in the treated group than the control group on day 18.

## 4. Discussion

After the harvest, goji fruits are prone to *A. alternata* infection, which leads to black mold rot [31] that not only harms human health but also causes economic losses. MLT application confers biotic stress resistance against pathogenic fungal infection, inhibits lesion expansion, represses pathogen development, and attenuates postharvest decay [32]. The current study showed that exogenous MLT treatment reduced disease incidence and lesion expansion in goji fruit during postharvest storage. As a potential antifungal agent, exogenous MLT has been shown to play positive roles in combating pathogens and restricting the postharvest decay in grapes [33], blueberry fruit [25], papaya [34], cherry tomato [35], and apple fruit [36]. Exogenous MLT application to postharvest fruit enhances the content of endogenous MLT, which, in turn, regulates many physiological changes in the fruit, including quality changes, senescence deterioration, and stress tolerance [20]. According to Yan et al. [35], exogenous MLT treatment helped retain the quality of cherry tomatoes at room temperature by increasing endogenous MLT. These findings corroborated the results of the current study showing that exogenous MLT treatment promoted the synthesis of endogenous MLT. Meanwhile, the increase in endogenous MLT level may be involved in the modulation of postharvest decay of goji fruit. However, the relationship between increased endogenous melatonin content and disease resistance needs further research.

The phenylpropanoid pathway is a vital part of secondary metabolism, functioning as a defense response by meditating the accumulation of metabolites that can resist pathogen invasion and infection [37]. The enzyme PAL plays a crucial role in limiting the rate of the pathway, followed by sequential catalyzation with C4H and 4CL to produce coumaric acid and *p*-coumaroyl-CoA that act as crucial precursors for numerous phenylpropanoid products like flavonoids and lignin [25]. CHS and CHI catalyze the first two steps in the synthesis branch of flavonoids. CAD is the key enzyme in lignin synthesis bypass to produce monolignols [7]. In addition to acting as an antioxidant enzyme to scavenge ROS, POD also plays a vital role in the process of polymerizing monolignols to form lignin [7,33]. MLT acts as a bioregulator during biotic and abiotic stresses, activating and/or modulating metabolic pathways, related hormones, and key elements that act against stressors [15]. In the current study, the activities of POD, PAL, C4H, 4CL, CHS, CHI, and CAD, and the expressions of *PAL*, *C4H*, *4CL*, *CHS*, *CHI*, and *CAD* in goji fruit were enhanced after exogenous MLT treatment during storage. Our results coincided with the findings of Qu et al. [25], who reported that MLT induced the activities and related genes of defense-related enzymes (POD, PAL, C4H, 4CL, and CAD), thus strengthening disease tolerance in postharvest blueberry fruit. In postharvest plum fruit [37] and papaya [34], MLT-induced increase in POD, PAL, C4H, and 4CL activities was found to be associated with reduced anthracnose and retention of fruit quality. As reviewed by Li et al. [33] and Arnao and Hernández-Ruiz [15], MLT treatment induces phenylpropanoid metabolism in the fruits by elevating the activities and expressions of phenylpropanoid pathway-related enzymes (e.g., PAL and POD), which in turn improves resistance against pathogens.

Secondary metabolites are compounds that help plants cope with various biotic and abiotic stresses and interact with positive and negative ecological factors. These metabolites are grouped into four classes based on their biosynthesis origin: terpenoids, polyketides, phenylpropanoids, and alkaloids [27,38]. Phenylpropanoids synthesized via the phenylpropanoid pathway, such as flavonoids, phenolics, lignin, lignans, and anthocyanins, comprise a crucial class of plant secondary metabolites containing more than 8000 aromatic compounds, which are primarily involved in plant development, ROS scavenging, protection against pathogen and herbivore, abiotic stress resistance, and so on [27,39]. In the present work, ten secondary metabolites in goji fruits were investigated. Based on their abundance on day 0, these metabolites, arranged from high to low abundance, were rutin (23,358 ng g^−1^), p-coumaric acid (3222 ng g^−1^), chlorogenic acid, ferulic acid, caffeic acid, sinapic acid, naringenin, quercetin, kaempferol, and protocatechuic acid. These findings were similar to the profiling and contents of wolfberry secondary products reported by Zhang et al. [7]. Further, our study showed that exogenous MLT treatment retained higher levels of caffeic acid, chlorogenic acid, ferulic acid, *p*-coumaric acid, protocatechuic acid, naringenin, quercetin, rutin, and kaempferol during storage compared to the control fruits. However, the treated and untreated fruits did not exhibit any significant differences in the sinapic acid levels. Secondary metabolites can be induced by various stresses and the application of exogenous elicitors [7,26]. MLT, as a master plant regulator, stimulates secondary metabolism by inducing the generation of anthocyanin, flavonoids, carotenoids, and other secondary compounds to regulate plant growth and provide resistance to biotic and abiotic stresses [15]. Analogous conclusions can be drawn from a previous study that reported that the application of exogenous MLT effectively controlled the postharvest decay of blueberry fruit by promoting the accumulation of disease resistance-inducing compounds, such as polyphenols, flavonoids, anthocyanins, and lignans [25]. Similarly, the MLT-treated litchi fruit [16], grapes [33], plums [37], cherry tomato [17], and papaya [34] exhibited higher levels of total phenols, flavonoids, anthocyanins, and lignin, which might improve the resistance to pathogenic fungi and restrict pathogen infection in fruit during cold storage. A recent study demonstrated that MLT treatment boosted the total phenolics and anthocyanin content and enhanced the antioxidant potential of blueberry fruit, which led to delayed postharvest decay [40]. Therefore, MLT can elicit high levels of polyphenols, flavonoids, and lignin and thus prevent the fruit from pathogen invasion and infection [18]. Notably, in our previous study, the levels of ·O_2_^−^ and H_2_O_2_ in 50 μmol L^−1^ MLT-treated goji fruit were higher during the first 6 days of storage and then lower than those in the untreated goji fruit. In contrast, the treated fruits always retain higher total phenols and flavonoids compared to the control fruits [28]. Some studies have shown that exogenous MLT induces ROS burst as the first cellular signal, followed by inducing MLT biosynthesis to boost endogenous MLT content at the beginning of biotic and abiotic stresses. Furthermore, endogenous MLT, as a positive effector against stresses, alters the expression of many genes and the modulating factors that alleviate or reverse the negative effects of pathogenic fungi on the physiological processes of plants [14,17,18]. Together, these results showed that MLT treatment restricted disease development in goji fruit postharvest and that MLT action was associated with ROS regulation and secondary metabolite generation.

Based on our previous and present study, we propose a possible mechanism mode of MLT on controlling the postharvest rot in goji fruit (Figure 6). The application of exogenous MLT in postharvest goji fruit induced an oxidation burst ($O_{2}^{\overset{-}{\cdot }}$ and H_2_O_2_), stimulated an increase in endogenous level of MLT, and thus accomplished the activation of disease response-related systems, that is, the antioxidant system (increase in enzymatic activities of and expressions of genes encoding PAL, C4H, 4CL, CHS, CHI, and CAD; increase in the levels of nine secondary metabolites including caffeic acid, chlorogenic acid, ferulic acid, *p*-coumaric acid, protocatechuic acid, naringenin, quercetin, rutin, and kaempferol), which was responsible for imparting pathogen resistance.

## 5. Conclusions

In conclusion, treating goji fruit with MLT at different concentrations has been shown to effectively reduce disease severity caused by *A. alternata* during storage. Additionally, 50 μmol L^−1^ of MLT treatment has been found to significantly increase endogenous melatonin levels in goji berries compared to the control group. Moreover, the activities of disease resistance-related enzymes containing POD, PAL, C4H, 4CL, CHS, CHI, and CAD were observed to increase to varying degrees, and the gene expressions related to defense reactions including *PAL*, *C4H*, *4CL*, *CHS*, *CHI*, and *CAD* were activated by MLT treatment. In addition, MLT treatment increased levels of secondary metabolites such as caffeic acid, chlorogenic acid, ferulic acid, *p*-coumaric acid, protocatechuic acid, naringenin, quercetin, rutin, and kaempferol. However, further exploration is required to provide direct evidence of the relationship between endogenous melatonin content and fruit defense response.

## Figures and Tables

**Figure 1 foods-12-04326-f001:**
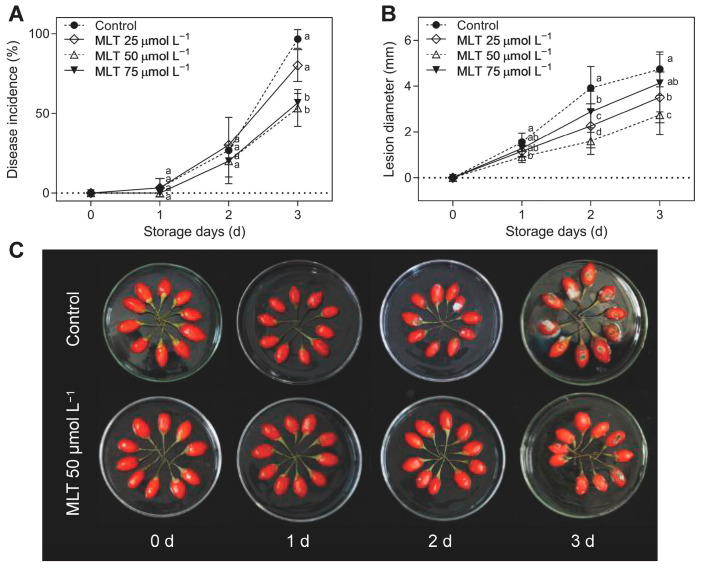
Disease incidence (**A**), lesion diameter (**B**), and visual appearance (**C**) in the untreated (control) and melatonin-treated (25, 50, and 75 μmol L^−1^) goji berries during storage at 28 °C; data are expressed as mean ± SD. The lowercase letters are used at each sample point to indicate a significant difference (*p* < 0.05) between control and melatonin-treated fruits.

**Figure 2 foods-12-04326-f002:**
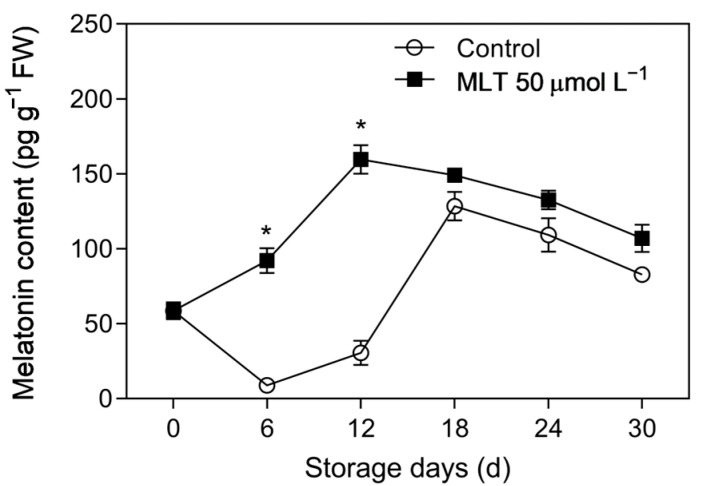
Endogenous melatonin content in the untreated (control) and 50 μmol L^−1^ melatonin-treated (the most effective treatment) goji berries during storage at 4 ± 0.5 °C. Data are presented as mean ± SD. “*” at each sample point indicates a statistically significant difference (*p* < 0.05) between control fruits and melatonin-treated fruits.

**Figure 3 foods-12-04326-f003:**
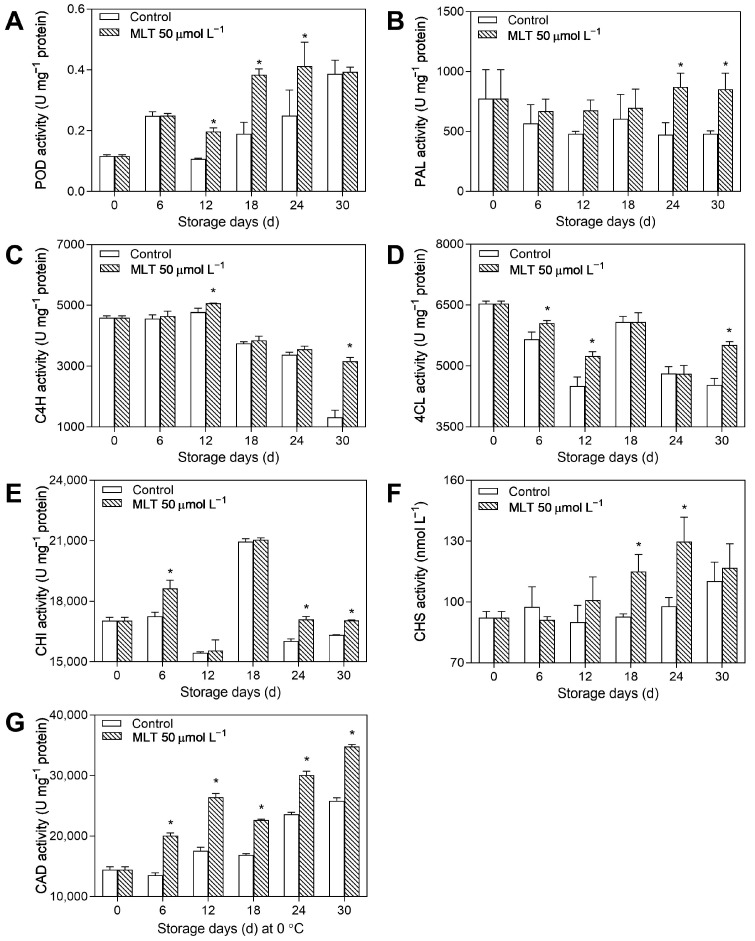
The activities of POD (**A**), PAL (**B**), C4H (**C**), 4CL (**D**), CHS (**E**), CHI (**F**), and CAD (**G**) in the untreated (control) and 50 μmol L^−1^ melatonin-treated goji berries during storage. Data are presented as mean ± SD. “*” at each sample point indicates a statistically significant difference (*p* < 0.05) between control fruits and melatonin-treated fruits.

**Figure 4 foods-12-04326-f004:**
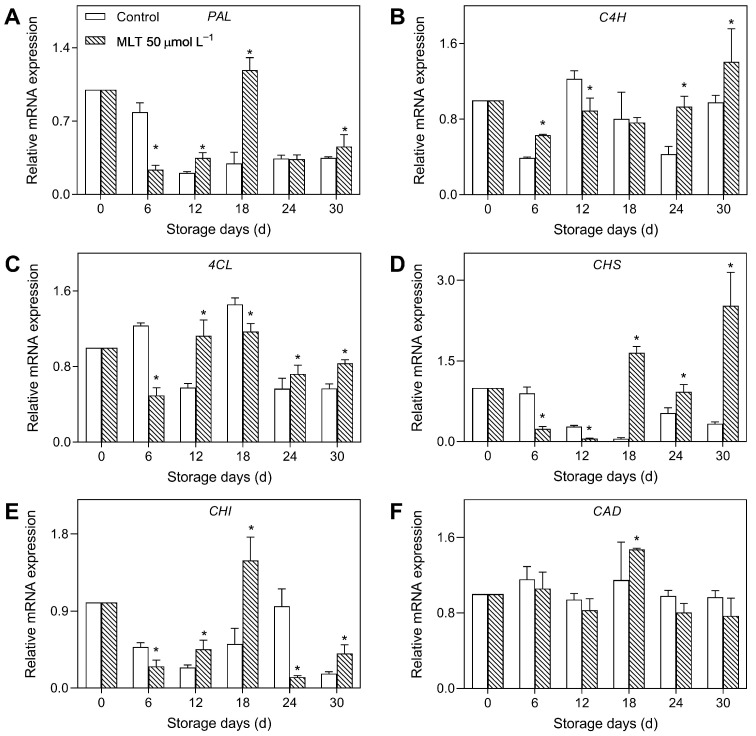
The expressions of *PAL* (**A**), *C4H* (**B**), *4CL* (**C**), *CHS* (**D**), *CHI* (**E**), and *CAD* (**F**) in the untreated (control) and 50 μmol L^−1^ melatonin-treated goji berries during storage. Data are presented as mean ± SD. “*” at each sample point indicates a significant difference (*p* < 0.05) between control and melatonin-treated fruits.

**Figure 5 foods-12-04326-f005:**
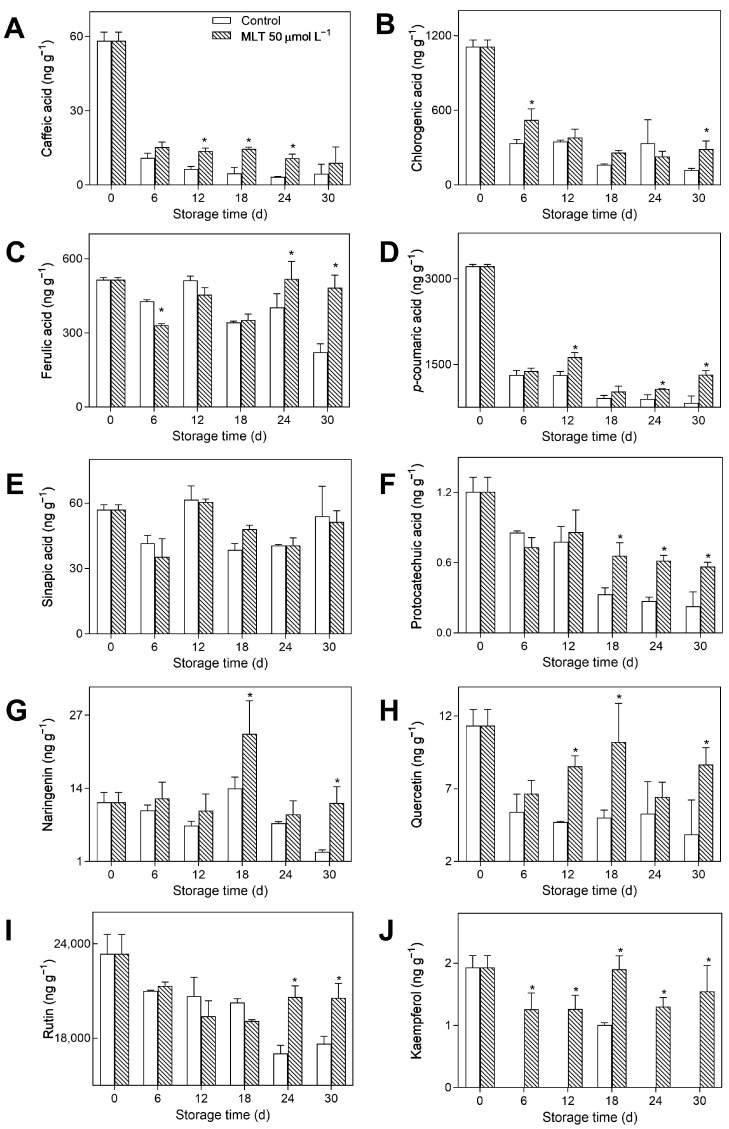
The contents of caffeic acid (**A**), chlorogenic acid (**B**), ferulic acid (**C**), *p*-coumaric acid (**D**), sinapic acid (**E**), protocatechuic acid (**F**), naringenin (**G**), quercetin (**H**), rutin (**I**), and kaempferol (**J**) in the untreated (control) and 50 μmol L^−1^ melatonin-treated goji berries during storage. Data are presented as mean ± SD. “*” at each sample point indicates a significant difference (*p* < 0.05) between control fruits and melatonin-treated fruits.

**Figure 6 foods-12-04326-f006:**
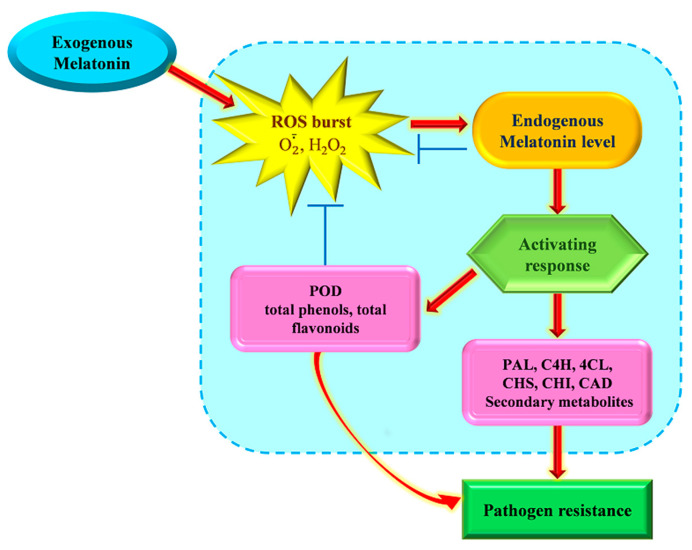
Mechanism of action of melatonin underlying postharvest decay of goji fruit during storage. The arrows and lines with bars indicate positive and negative regulatory actions. POD, peroxidase; PAL, phenylalanine ammonia-lyase; C4H, cinnamate 4-hydroxylase; 4CL, 4-coumarate-CoA ligase; CHS, chalcone synthase; CHI, chalcone isomerase; CAD, cinnamyl alcohol dehydrogenase.

**Table 1 foods-12-04326-t001:** Primers used in the qRT-PCR analysis.

Gene Name	Forward Primer (5′ to 3′)	Reverse Primer (5′ to 3′)	Product (bp)
*PAL*	CGCCACGAGATAGGTTGATGACAT	GGCTTCTTACTGCTCGGAACTTCA	169
*C4H*	GCAAGTGACCGAGCCAGACA	AGCCACCAAGCATTAACCAAGATC	177
*4CL*	TGAGAGTCGGAGCAGCGATATTGA	TTCCTAATGGAGCACCACCAGACA	195
*CHS*	CACTGATAGCGAGCACAAGACTGA	AGGCACCTCAACAACCACTATGTC	180
*CHI*	CTGCCACTGCAAAGATTTCATCCA	CCCTGCTTCCACCTAAGTACCATT	114
*CAD*	ACCAGAACAGGCAGCACCACTA	CTCCCATGTGTCCAACTCCTCCT	128
*Actin*	CTCAGCACCTTCCAGCAGAT	TAACACTGCAACCGCATTTC	162
*EF1α*	GAAGGGTGTCCCTCAGATCA	CCGTCCATGTCGTCTCTTTT	180

## Data Availability

All the data presented in this study are available within the article.

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
