# Peer review of "Metabolic Profiling and Molecular Mechanisms Underlying Melatonin-Induced Secondary Metabolism of Postharvest Goji Berry (Lycium barbarum L.)"

_foods, 2023, doi:10.3390/foods12234326_

Round 1

Reviewer 1 Report

Comments and Suggestions for Authors

Dear Authors, the manuscript should be revised according to the minor comments highlighted across the text.

Reviewer 2 Report

Comments and Suggestions for Authors

Manuscript ID foods-2729995-peer -review

Metabolic profiling and molecular mechanisms underlying melatonin-induced secondary metabolism of postharvest goji berry (Lycium barbarum L.)

Junjie Wang, Huaiyu Zhang, Jie Hou, En Yang, Lunaike Zhao, Yueli Zhou, Wenping Ma, Danmei Ma and Jiayi Li

Preserving the quality of fresh fruits of food and medicinal plants is important for crop production, food and pharmacological industries. Currently, there are many traditional technologies for long-term preservation of plant products after harvesting. It is necessary to study the mechanisms of resistance to postharvest stress to justify the available methods and to develop alternative methods.

Melatonin as a polyfunctional compound can delay postharvest senescence and preserve the quality of goji fruit. The relevance of this study is related to deciphering the mechanism of fruit resistance to postharvest stress. The authors studied the dynamics of endogenous melatonin in harvested fruits, the activity of several enzymes, and screened gene activity and secondary metabolite content. They evaluated the effect of melatonin on resistance of goji fruits to pathogens and hence prevention of fruit rot.

There are comments on the text of the manuscript.

1.    Methodology line 124 and Results. fig. 2. The authors determined the content of endogenous melatonin in fruit. Why is the concentration of this substance in Figure 2 expressed in ng L-1 (this unit of measurement for the solution in the well of the plate in ELISA analysis). It is necessary to convert the melatonin content to the weight of the berry.

2. Methodology. The authors incubated goji fruits for 5 min in melatonin solution. Since the size of the berries and, above all, the surface area are not given, it is difficult to estimate how much melatonin can enter the fruit? Therefore, it is difficult to recommend this method in practice.

3. Methodology. The authors did not specify the drying time of the fruit after melatonin treatment. Therefore, it is impossible to estimate the additional time of melatonin absorption through the peel.

4. Methodology. What effect did treatment with 75% ethyl alcohol solution have on peel integrity? Alcohol has a permeability through membranes. What is its effect on the fruit cuticle? Does it reduce pathogen resistance in controls?

5. Methodology. The authors evaluated the efficacy of melatonin on pathogen resistance at 28 degrees Celsius and fruit storage at 4 degrees Celsius. Can different conditions (different temperature) be considered as the norm for evaluating the efficacy of the same concentration of melatonin and the existence of the same mechanism of action of melatonin.

6. Results. Lines 228 and 265In Figures 2 and 3, the sign of differences between mean parameters is represented by an asterisk (*) rather than a lowercase letter as indicated in the captions of these figures.

7. Results. Lines 287-288. It is not clear which parameters are being compared in Figure 4. The caption to Figure 4 indicates that the parameters of control fruit and melatonin-treated fruit are compared. In this case, the difference sign (*) should only be at the MT 50 µmol L-1 columns.

8. References. The authors of the manuscript cited 7 of their own papers: 3, 6, 7, 26, 29, 30,33.

9. Discussion. Line 315-316. The authors' assumption that increased endogenous MLT levels contributed to the modulation of postharvest decomposition of goji fruit is not generalized. Apparently, this is due to the lack of correlation analysis and established relationships between physiological and biochemical parameters.

Question on environmental safety of wide application of melatonin in food industry.

8. The use of melatonin increases the amount of melatonin in berries. Whether accumulation of melatonin in fruits is not a danger to humans. How much melatonin-treated goji berries can be consumed, as it is known that melatonin in large amounts and during daylight hours (daylight hours) can alter a person's circadian rhythms and also affect fertility. 

Reviewer 3 Report

Comments and Suggestions for Authors

The article entitled “Metabolic profiling and molecular mechanisms underlying 2 melatonin-induced secondary metabolism of postharvest goji berry (Lycium barbarum L.)” submitted to the journal presents a study aimed to determination the impact of MLT on goji decay induced by Alternaria alternata during storage and explore the MLT-mediated modulation of secondary metabolism and metabolic profiling to identify the potential underlying defense mechanisms.

The search for new natural-like products to protect different crops, fruits and food against filamentous fungi is in line with current trends in agriculture and food processing.

The manuscript is clearly structured and written in a language understandable to potential readers. The chosen methodology meets the stated objectives of the study. The results obtained have been satisfactorily presented and discussed.

I have the following remarks on the article:

L 90 - should it be goji berry?

L 96 - goji berries - were they L. barbarum or L. chinese?

L 383 – 396 - I recommend reconsidering the content of the conclusion and moving the proposed scheme (Fig. 6) into the discussion. Conclusion should not reiterate the results obtained; it should answer the aim of the present study. Conclude what new findings were found and how the findings advance our knowledge in the field.

There are some minor incorrectness in the text, the authors should re-read the text carefully.

The obtained findings are important for further research and future application. This research is in line with current scientific trends.
